# Distribution of Denitrification among Haloarchaea: A Comprehensive Study

**DOI:** 10.3390/microorganisms9081669

**Published:** 2021-08-04

**Authors:** Jose María Miralles-Robledillo, Eric Bernabeu, Micaela Giani, Elena Martínez-Serna, Rosa María Martínez-Espinosa, Carmen Pire

**Affiliations:** 1Biochemistry and Molecular Biology Division, Agrochemistry and Biochemistry Department, Faculty of Sciences, University of Alicante, Ap. 99, E-03080 Alicante, Spain; jmmr19@alu.ua.es (J.M.M.-R.); ebs40@alu.ua.es (E.B.); micaela.giani@ua.es (M.G.); elenamrtnz16@gmail.com (E.M.-S.); rosa.martinez@ua.es (R.M.M.-E.); 2Multidisciplinary Institute for Environmental Studies “Ramón Margalef”, University of Alicante, Ap. 99, E-03080 Alicante, Spain

**Keywords:** haloarchaea, *Halobacteriaceae*, *Haloferacaceae*, *Natrialbaceae*, *Halorubraceae*, denitrification, respiratory nitrate reductase, respiratory nitrite reductase, nitric oxide reductase, nitrous oxide reductase

## Abstract

Microorganisms from the *Halobacteria* class, also known as haloarchaea, inhabit a wide range of ecosystems of which the main characteristic is the presence of high salt concentration. These environments together with their microbial communities are not well characterized, but some of the common features that they share are high sun radiation and low availability of oxygen. To overcome these stressful conditions, and more particularly to deal with oxygen limitation, some microorganisms drive alternative respiratory pathways such as denitrification. In this paper, denitrification in haloarchaea has been studied from a phylogenetic point of view. It has been demonstrated that the presence of denitrification enzymes is a quite common characteristic in *Halobacteria* class, being nitrite reductase and nitric oxide reductase the enzymes with higher co-occurrence, maybe due to their possible role not only in denitrification, but also in detoxification. Moreover, copper-nitrite reductase (NirK) is the only class of respiratory nitrite reductase detected in these microorganisms up to date. The distribution of this alternative respiratory pathway and their enzymes among the families of haloarchaea has also been discussed and related with the environment in which they constitute the major populations. Complete denitrification phenotype is more common in some families like *Haloarculaceae* and *Haloferacaceae*, whilst less common in families such as *Natrialbaceae* and *Halorubraceae*.

## 1. Introduction

Saline and hypersaline environments are found worldwide. Hypersaline environments can be divided into two groups: thalassohalines, derived from seawater and containing Na^+^ and Cl^−^ as predominant ions, and athalassohalines, which display variable ion composition depending on the area where these environments develop [1,2]. The biodiversity of these ecosystems depends on several factors such as total salinity and ionic composition, pH, temperature, radiation, water accessibility and oxygen availability. In terms of respiration, due to lower oxygen solubility in these environments, some microorganisms have developed different strategies. Some examples are the production of gas vesicles (also called gas vacuoles) that enable cells to float to the surface of the brines where oxygen concentration is higher or the use of N-compounds as final electron acceptor, such as nitrate or nitrite using denitrification pathway [3,4].

Microbial communities play relevant roles in biogeochemical cycles in nature. One of them is the N-cycle, the conversion of nitrates into gaseous nitrogen-containing compounds and vice versa, which is of particular importance in connection with human anthropogenic activities. It is worth noting that anthropogenic activities such as agriculture contribute to the increase of nitrate, nitrite, and ammonium. This, in turn, leads to the fact that most of the nitrous oxide (N_2_O) is released by microorganisms after the transformation of the reactive nitrogen in fertilized agricultural zones [5,6,7]. N_2_O is a potent greenhouse gas, and it contributes greatly to the depletion of the ozone layer [8,9,10]. In this regard, denitrification is noteworthy, because it is one of those metabolic pathways that is not only the major source of biological N_2_O, but also the only known sink, which allows one to mitigate N_2_O emissions [2]. Therefore, microbes showing metabolic capabilities related to N-cycle are worth investigating.

Denitrification is an alternative respiratory process in the nitrogen cycle in which nitrate (NO_3_), nitrite (NO_2_), nitric oxide (NO) and nitrous oxide (N_2_O) are reduced successively to nitrogen gas (N_2_) thanks to the activity of a set of four enzymes: nitrate reductase (Nar), nitrite reductase (Nir), nitric oxide reductase (Nor) and nitrous oxide reductase (Nos) [11]. The respiratory nitrate reductase catalyses the first reaction in which the nitrate is reduced to nitrite. In many haloarchaea, this enzyme contains a short N-terminal sequence known as twin-arginine translocase motif (TAT). The TAT system transports folded proteins, that contain this motif, across the plasma membrane, and it plays an important role in protein transport in haloarchaea locating the enzymes in the periplasm, on the membrane potential-positive side of the cytoplasmic membrane as in the case of Nar [12]. Bioenergetically, respiratory nitrate reductases from haloarchaea are classified as pNar. The electron transport from Q-pool to the molybdenum active site Nar, via cytochrome b/Rieske protein complexes and iron-sulphur clusters, is poorly described in these microorganisms [13]. Nitrite reductases (Nir) catalyses the second denitrification reaction, reducing nitrite to nitric oxide. There are two main types: copper-containing nitrite reductases (NirK) and cytochrome-*cd1*-dependent nitrite reductases (NirS). NirK are homotrimers that contain two copper sites (Type 1 and Type 2) per subunit, whereas NirS shows a homodimeric structure and contains haems *c* and *b1* as cofactors [14]. The next step of denitrification process is catalysed by nitric oxide reductases (Nor). There are three main subtypes of Nor enzymes: short-chain respiratory NORs (scNORs or cNORs) that contain a catalytic subunit (NorB) and an integral membrane subunit (NorC); long-chain respiratory NORs (lcNORs or qNORs) with a unique NorZ subunit and copper NORs (CuANORs) whose di-copper site (CuA) holds the reduction of NO. All three types present one haem *b* and a binuclear metal centre consisting of haem *b3* and FeB [15,16,17]. In the case of haloarchaea, NorZ enzymes, are widely distributed being, in fact, the only class of nitric oxide reductase found in these microorganisms [18]. Nitrous oxide reductase (Nos) catalyses the last step of denitrification. This enzyme has been extensively studied during the last twenty years in bacteria, particularly from soils, due to the relationship of nitrous oxide with climate change [19]. Interestingly, some bacteria genomes have *nosZ* genes, which are responsible for the reduction of N_2_O to N_2_, while lacking the rest of denitrification genes [20]. Previous research indicates that they might consume the N_2_O released by other microorganisms during denitrification [21]. Furthermore, *nosZ* gene can also be absent in some species; thus, generating N_2_O as final product [11]. Recently, a causal relation has been reported between the composition of the denitrifier community in soil and the N_2_O emissions caused by changes in the proportion of those denitrifiers, whose genome has *nosZ* gene [22]. Therefore, the co-occurrence of the denitrification genes plays an important role in the determination of the potential N_2_O emission from a given ecosystem [23]. Although denitrification has been extensively studied in several bacterial species, there is scarce information about haloarchaeal denitrifiers. Interestingly, haloarchaea constitute the prevailing population in hypersaline environments and their activity contribute importantly to biogeochemical cycles. At the time of writing this work, most species included in *Halobacteria* class have been described as denitrifiers [24,25]. However, it remains open to question whether they are complete or partial denitrifiers. For this reason, analysing the distribution of denitrification genes in haloarchaea might contribute to the understanding of the role of these microorganisms in nitrogenous gasses release related to global warming. Moreover, extremophilic complete denitrifiers have aroused interest of researchers and industries in search of innovative ideas and processes for wastewater bioremediation [25,26,27]. Most evidence suggests that not all denitrifying microorganisms possess the complete set of enzymes required to perform all steps within the pathway; and therefore, they conduct only a subset of the involved enzymatic reactions which makes difficult and even impossible their application to efficient wastewater treatments [28]. For this purpose, this research displays for the first time the phylogenetic distribution of denitrification in haloarchaea members using bioinformatic tools and a discussion about the evolution of denitrification in these microorganisms.

## 2. Materials and Methods

### 2.1. Data Sample and Haloarchaeal Phylogenetic Tree Constructing

Selection of the16S rRNA gene sequences from haloarchaea species included in this study was carried out using the RefSeq database of NCBI (https://www.ncbi.nlm.nih.gov/refseq; accessed 15 January 2021) [29,30,31].

All the 16S rRNA gene sequences available from haloarchaea at RefSeq (147 sequences) were used to construct a phylogenetic tree to study distribution patterns of the four main enzymes of denitrification (Nar, Nir, Nor, Nos). For this purpose, these sequences were aligned using the MUSCLE algorithm [32]. Afterward, a phylogenetic neighbour-joining tree (Boostrap 1000) was built with these alignments using MEGA-X software (https://www.megasoftware.net; accessed on 15 January 2021) [33]. Annotation of the phylogenetic tree was accomplished with the online tool “Interactive Tree Of Life (iTol) v4” (https://itol.embl.de; accessed on 15 January 2021) [34]. Taxonomic classifications are according to the Genome Taxonomy Database (GTDB) (https://gtdb.ecogenomic.org; accessed on 15 January 2021) and to NCBI Taxonomy (https://www.ncbi.nlm.nih.gov/taxonomy; accessed on 15 January 2021) [31,35,36,37].

### 2.2. Search for Denitrification Genes and Proteins

For each species, manual and automated searches of proteins and genes encoding the four main enzymes of denitrification have been performed. NCBI Protein (https://www.ncbi.nlm.nih.gov/protein; accessed on 15 January 2021), NCBI Gene (https://www.ncbi.nlm.nih.gov/gene; accessed 15 January 2021) and Integrated Microbial Genomes & Microbiomes system v.5.0 (IMG/M) (https://img.jgi.doe.gov; accessed on 15 January 2021) databases were used for that purpose [31,38]. Firstly, the manual search has been accomplished using the search equations displayed in Table 1. Secondly, amino acid sequences from well-described enzymes retrieved with the manual search, were utilized as query for a BLAST automatic search (IMG/M BLAST tool; default parameters used). Enzymes found with at least 65% identity were recovered as good match. Subsequently, to avoid possible errors, a manual revision of each protein retrieved was accomplished examining the genetic context and domains by means of the bioinformatic tool NCBI Sequence Viewer 3.33.0 (https://www.ncbi.nlm.nih.gov/tools/sviewer; accessed on 15 January 2021) and the Integrated Microbial Genomes & Microbiomes system v.5.0 database (https://img.jgi.doe.gov; accessed on 15 January 2021) [38]. For nitrate reductase, only the catalytic subunit NarG has been examined, but in manual searching the presence of an adjacent NarH subunit coding gene has also been verified. All these data were integrated and classified in a local database for their analysis. In this study the term “denitrifier” is used for those species with at least one of the four main enzymes of denitrification encoded in their genome and the term “nondenitrifier” for microorganisms with available genome in which no genes encoding any of the main enzymes of denitrification was found. Thus, it has been assumed that species whose genomes only show genes coding for one of the four main enzymes, may not be proper denitrifiers based on canonical definitions, although they could contribute on denitrification as part of mixed populations in natural environments. Moreover, based only on the genomic data, it is not possible to verify if the genes are expressed and consequently, the microorganism can denitrify, therefore these denitrifiers should be considered as potential denitrifiers in the context of this work.

## 3. Results and Discussion

### 3.1. Data Sample and Taxonomic Diversity. How Widespread Is Denitrification in Haloarchaea?

One hundred and forty-seven haloarchaea species were retrieved from the RNA 16S gene search. These species have been taxonomically classified in 44 genera and 10 families to look for denitrification patterns at these taxonomic levels (Table 2). From the analysis of this classification, it can be assumed that the number of available genomes from *Halobacteria* class is suitable and significant for this research. There are representatives from all described families in literature, and from the great part of the genera [35,36,37]. Nevertheless, due to the low number of species classified in each genus, it was concluded that genera classification is not suitable for a deeper study. Hence, all subsequent analysis has been focused on the family taxon.

The main objective of this research is not only to clarify the role of haloarchaea in denitrification, but also to deepen in their contribution to climate change due to the production of nitrogenous gases through denitrification. For this reason, after compiling current data about the classification of these microorganisms, the next step was to study how genes encoding enzymes of denitrification were distributed among them, for a further classification as complete or partial denitrifiers. From the initial 147 identified species (16S rRNA gene sequences), only 122 genomes were available (draft or complete; consulted in January 2021). Thus, these 122 genomes have been analysed to inspect for the presence of the four main enzymes of denitrification. With the aim of facilitating this analysis, all these data were integrated in a single phylogenetic tree (Figure 1a) and arranged in Table 3.

Considering data from the global phylogenetic tree and the distribution of denitrification in *Halobacteria,* denitrification can be assumed to be a common pathway in this class of microorganism (this issue is deeply discussed in the next sections). From the 122 species with accessible genome, only 36 do not contain genes that encode any of the four main enzymes involved in denitrification, whereas the rest, at least encode one of them (NarG, NirK, Nor or Nos). Figure 1b displays the distribution of the four main enzymes of denitrification in haloarchaea, being the nitrite reductase and nitric oxide reductase the most widespread and, conversely, nitrate reductase and nitrous oxide reductase the less.

As can be seen in Figure 1a, some species lack for the genes encoding NarG and NarH subunits according to our genome analysis, even though they have been proved to be capable of growing under anaerobic conditions using nitrate as electron acceptor. These species are *Halobacterium jilantaiense, Halobacterium salinarum, Halorhabdus tiamatea, Haloterrigena saccharevitans, Natrinema altunense, Natrinema versiforme* and *Salinirussus salinus* [39,40,41,42,43,44,45]. Exploring in detail their genomes, it has been found that they contain sequences coding for other molybdopterin oxidoreductases showing low sequence identity with NarG and NarH. Besides, in some of these species, the β-subunit analogous to NarH, usually encoded in the same operon, was not found. This ability of nitrate reduction under anoxia in the absence of a canonical respiratory nitrate reductase may be explained by the possibility that other molybdopterin oxidoreductase encoded in their genomes could use nitrate as an electron acceptor, as occurs in some enzymes of the DMSO reductases family, which are able of catalyse redox reactions in the presence of different electron acceptors as in the case of nitrate reductase and perchlorate reductase [27,46,47]. Nevertheless, this enzymatic capability must be monitored by in vivo/in vitro enzymatic analysis. It is also interesting that the six NarG sequences found in the *Natrialbaceae* family are longer than those NarGs found in the other haloarchaeal genomes. However, bioinformatic analysis of these sequences did not reveal any differences or extra domain compared to NarG sequences described from other haloarchaeal members. Furthermore, molybdenum cofactor binding site as well as Fe-S cluster binding sites are very well conserved in these NarGs (data not shown).

In the case of nitrite reductase, the most remarkable aspect is that all the retrieved sequences code for copper-nitrite reductases type (NirK); consequently, none of them are cytochrome *cd_1_*-nitrite reductase type (NirS). Interestingly, many previous studies investigating the distribution of *nirS* and *nirK* along different ecosystems indicate that environmental conditions such as low oxygen availability, high water content or high salinity are favourable for the *nirS* carrying communities [23,48,49,50,51]. Furthermore, in these studies it is also pointed out that *nirK* gene is not even detectable in samples at “high salinity” [49,52,53]. However, it is worth highlighting that these studies are neither carried out nor focused on hypersaline environments, such as salt marshes or salty ponds, whose microbial communities differ from other less saline ecosystems and in which the *Archaea* domain populations are predominant [2,54,55]. Despite this, our data reveal that NirK is the dominant class of nitrite reductase in haloarchaea, as well as qNor was reported as the main type of nitric oxide reductase in these extremophilic microorganisms (despite *nor* gene is sometimes misannotated in haloarchaea as *cd*Nor all of them are qNor) [18]. About NosZ, all the sequences from haloarchaea retrieved in this study belonged to the Clade I group of “typical” Nos characterized by the presence of the TAT-signal peptide.

### 3.2. Partial vs. Complete Denitrifiers. Distribution by Family Taxon

The advantage of identifying the presence or absence of each of the main genes of denitrification in the taxonomic tree is that it allows to identify common patterns relative to the denitrification capabilities in each family. Even considering that a significant number of the families are not well represented *(Halobacteriaceae, Haloccocaceae, Natronoarchaceae, Salinarchaceae, Haladaptataceae* and *Halalkalicoccaceae)* because of the low number of species, well-represented families (*Natrialbaceae, Haloarculaceae, Halorubraceae* and *Haloferacaceae*) can provide an accurate view about the predominance of complete or partial denitrification among haloarchaea. Table 3 and Figure 2 display the data arranged by family.

Denitrification is a widespread respiratory pathway in haloarchaea. 70.5% of the reviewed species show potential capability for denitrification being 58.2% partial denitrifiers and 12.3% of them complete denitrifiers. These percentages are similar to other studies performed in soils and are directly connected to the idea that a common complete reduction can be driven by the contribution of several partial denitrifiers [56,57]. The high percentage of partial denitrifiers indicates that complete denitrification seems not to be a major process in hypersaline environments. However, the fact that the majority of the species encode for at least one and more than a half for at least two key enzymes involved in denitrification states how important this process is in ecosystems with low oxygen availability [58,59,60,61].

Regarding denitrifying communities, differences among the well-represented families can be established. *Haloarculaceae* and *Haloferacaceae* are the families comprising most potential denitrifying species. Meanwhile, *Natrialbaceae* and *Halorubraceae* have more non-denitrifying species. One of the explanations about these differences among haloarchaea could be related to the characteristics of the environment they inhabit. A significant number of species that belong to the *Natrialbaceae* and *Halorubraceae* families are haloalkaliphiles. Haloalkaliphilic ecosystems are soda lakes and soda soils characterized by presenting extremely high salinity and pH (up to 11). The biological sulphur cycle is one of the most active in that kind of habitats and it is connected to many other cycles, such as carbon, nitrogen, or metal cycles [62,63,64,65]. Functional groups of fermenters and sulfidogens are the main anaerobic microorganisms in these ecosystems due to the availability of organic matter and oxidized sulphur species, whereas complete denitrifiers are scarce and denitrification does not have the same important role and abundance as in neutral ecosystems [64,65]. The accessibility to other alternative electron acceptors could explain the differences among families in denitrification patterns, but more research is needed to establish a relationship.

This relationship between type of environment and denitrification can also be examined with the co-occurrence of enzymes and frequency of complete/partial denitrification (Table 3 and Figure 2). On one hand, members from *Natrialbaceae* and *Halorubraceae* families are mostly non-denitrifiers or partial denitrifiers with 1–2 enzymes. In addition, complete denitrification was only identified in *Natrialbaceae* representing only 5% of the species. On the other hand, *Haloarculaceae* and *Haloferacaceae* members, generally have more complete denitrification pathways with 3-4 enzymes. Complete denitrification is more predominant in *Haloarculaceae*, and partial denitrification is more extended in *Haloferacaceae*.

Partial and complete denitrification is closely related to gas emissions. NO and N_2_O are gaseous products of the second and third reactions of denitrification and these two N-oxides have been extensively studied the last years due to their relationship with ozone layer deterioration and climate change [66,67,68]. NO participates in ozone layer depletion and is also considered a precursor of greenhouse gases, whereas N_2_O is a powerful greenhouse gas with an atmospheric heat-trapping ability 265 times higher than CO_2_ (considered over a 100-year period) [8,9,10]. Both are directly or indirectly related to ozone layer depletion and there is almost no information about their release in hypersaline environments. To shed some light on this topic, the capacity of haloarchaea to produce these gases has been examined, considering the presence of microorganisms with pathways that end with the formation of NO or N_2_O. The ability to mitigate greenhouse emissions by performing the last step of denitrification by Nos has also been discussed.

Figure 2 shows all the available possibilities identified. Denitrifiers whose theoretical end-product is NO are species that carry NarG-NirK or only NirK. These denitrifiers accomplish 12.8% of the total potential denitrifiers (9% of total available haloarchaeal genomes), while species whose pathways theoretically end in N_2_O (species that carry Nar-Nir-Nor, Nar-Nor, Nir-Nor or only Nor) make up the 46.5% of the potential denitrifiers (32.8% of total available haloarchaeal genomes). Therefore, 59.3% of potential haloarchaeal denitrifiers (48.6% of total available haloarchaeal genomes) can be potentially emitting greenhouse gases to the atmosphere. However, the ability of reduce greenhouse emissions are mainly carried out by complete or partial denitrifiers that possess Nos. *nos* gene appears in 39.5% of the denitrifiers genomes (16.5% of total available haloarchaeal genomes). This percentage in agreement with other studies, concluding that the absence of this gene in the majority of denitrifiers is not considered a major lost in terms of bioenergetic yield [69]. Considering that the frequency of NO and N_2_O potential releasers is higher than N_2_ releasers, and arid and semiarid ecosystems are expanding and increasing their salinity due to global warming and anthropogenic activities [70,71,72,73], denitrifying haloarchaea could be contributing to emissions of N-oxides more than it was initially expected [74]. Nevertheless, how these NO and N_2_O emissions can be mitigated by denitrifiers having *nor* and/or *nos* genes is still unknown. In addition, more research efforts are required to understand the possible synergy between microbial communities in which the final product of some partial/truncated pathways is used by other microorganisms.

### 3.3. Has Complete Denitrification Evolved from an Ancestral Respiration-Detoxification Function in Haloarchaea?

One of the currently unaddressed question marks related to denitrification in haloarchaea refers to its connection with two different processes: cellular detoxification (NO_2_^−^ and NO are toxic compounds) and anaerobic respiration (NO_3_^−^, NO_2_^−^ and NO as final electron acceptors instead of O_2_) [75,76]. Considering that both partial and complete denitrification have been reported in haloarchaea, the physiological/metabolic aims and the evolution of haloarchaea denitrification is worth exploring to elucidate whether this pathway evolved from a primitive respiratory process or from a pathway for detoxification [24,58].

To explore these hypotheses, the predominant pattern regarding the number of enzymes for denitrification encoded in haloarchaeal genomes was studied (Figure 2). The most frequent groups are those in which there are two or more enzymes catalysing subsequent steps of denitrification (NarG-NirK, NirK-Nor, Nor-Nos, NarG-NirK-Nor, NirK-Nor-Nos and NarG-NirK-Nor-Nos). Therefore, pathways with nonconsecutive steps are not common. The groups that at least involve NirK-Nor are the most numerous, followed by complete denitrifiers.

A great effort has been made to clarify how this metabolic pathway has evolved. However, incongruences with 16S rRNA gene sequences and gene annotations as well as with denitrification genes phylogeny have made this task very difficult [11,77]. Over the years, two main hypotheses about the evolution of this respiratory pathway have been postulated. Some studies about geochemical and oceanographic data have proposed that denitrification appeared after or at the same time as the great oxygenation of Earth [78,79]. Conversely, the most current studies based on genomics, microbial interactions with the environment, bioavailability of metals and (bio)geochemical fluxes, proposed the opposite. Denitrification should have appeared in the Archaean era, prior to aerobic respiration, even before stabilization of the independent domains of life and when nitrogen flux was still largely controlled by abiotic activities [80,81,82,83].

Regarding these studies, copper should not have played a role in the Archaean era because of its low bioavailability in this ancient environment. Thus, this metal was bioavailable after the great oxygenation [84,85]. This means that enzymes like NirK and NosZ appeared after the great oxygenation. Genome analysis has demonstrated that NirK is the only class of respiratory nitrite reductase in haloarchaea; therefore, it is unlikely to think about a denitrification pathway lacking the copper dependent enzymes NirK and NosZ. Hence, the appearance of denitrification in *Halobacteria* must have occurred after the great oxygenation event. Besides, the high frequency of co-occurrence of *nirK* and *nor* genes in haloarchaeal genomes is also linked with the idea that early bioenergetic systems should have evolved from detoxification pathways that inactivate reactive species. Related to these aspects Figure 1a was used to establish possible evolutive relationship between the enzymes and analyse this idea.

Considering that oxygen availability on hypersaline systems is low, it is not surprising that a significant number of haloarchaeal genomes encode for the full pool of denitrification enzymes. Complete denitrification is an adaptative advantage, which allow microorganisms to survive under stressful conditions like anoxia. Nevertheless, partial denitrifiers with only NirK and Nor enzymes in haloarchaea seem to be very frequent, in spite of the low bioenergetic yield. Genetic linkage of these two genes have been previously observed in haloarchaea and other taxonomic groups, but this is not a usual characteristic in denitrifying microorganisms [11,18,86]. However, our data shows that these two genes seems to be linked in haloarchaea due to their high frequency of co-occurrence and their proximity in the haloarchaeal chromosome [25]. The absence of *nar* or *nos* genes in that kind of partial denitrifiers can be in conjunction with the fact that a large group of haloarchaea have *nar* and *nos* operons in plasmids instead of being in the main chromosomes, what could indicate that these genes could be acquired later by horizontal gene transfer (HGT) [11,25]. Although in this research the phylogenetic origin of these two genes has not been investigated, one hypothesis is that this transference was from halophilic bacteria. The structure of Nos from haloarchaea is very similar to its bacterial counterpart, and Nar, although differs from the typical bacterial enzyme in the presence of the TAT signal and in its membrane orientation, structurally is also very similar to the bacterial Nar [87] These results highlight the importance of NirK and Nor enzymes as the initial core for respiration and/or detoxification of N-oxides in haloarchaea. We propose that the common ancestor of some of the current complete denitrifier haloarchaea carried only NirK and Nor. Furthermore, this denitrifying ancestor should have appeared after the Earth’s oxygenation due to the copper dependency of NirK. These two enzymes allowed this microorganism not only to detoxify N-oxides, but also to survive under microaerophilic or anaerobic conditions, helped by other microorganisms able of reducing nitrate to nitrite or by the abiotic reduction of nitrate to nitrite. This partial denitrification is likely to produce a low bioenergetic yield, assuming that only Nor (qNor) could generate proton-motive force, but information about the electrogenic role of qNors are still scarce [88,89]. Finally, in an evolutionary event, this partial denitrifier ancestor acquired the *nar* and/or *nos* genes by HGT, which allowed a more efficient denitrification pathway and a better adaptation to the environment.

## 4. Conclusions

Sequencing technology and genome databases have provided high quantity of data during the last decade, leading to the development of new bioinformatics tools [90]. This research aims the analysis of available haloarchaea genomes to provide a new insight on denitrification driven by haloarchaea.

Acquisition of genomes, taxonomic classification and subsequent enzyme annotation have shown that details on denitrification (and probably other metabolic pathways) at genus level in haloarchaea are still scarce, due to the lack of genomes from microorganisms belonging to the same taxonomic level. However, analyses at family level can be conducted with at least microorganisms from *Natrialbaceae*, *Haloferacaceae*, *Haloarculaceae* and *Halorubraceae* families.

This deep analysis has provided new insights for future haloarchaeal denitrification studies and has confirmed some hypothesis about this respiratory process in hypersaline environments. Enzyme-by-enzyme analysis has shown that NirK is the only type of respiratory nitrite reductase present in haloarchaea. These data state that the set of denitrification enzymes defined in haloarchaea are pNar, NirK, qNor and NosZ Clade I. Furthermore, denitrification genes have been demonstrated to be widespread on hypersaline environments, in which haloarchaea encompasses most of the microbial community. Genome analysis has also revealed that more than a half of the species are potential gas emitters that release NO or N_2_O to the atmosphere, causing hypersaline ecosystems to be considered as sources of greenhouse gases. This fact highlights the importance of putting more efforts in understanding the contribution of these ecosystems to climate change, and as it has been pointed out, interactions between denitrifiers should be addressed to understand better the flux of emissions.

Evolution of this pathway in haloarchaea is subject to controversy but copper dependency of denitrification enzymes suggests that haloarchaeal denitrification should have appeared after the great oxygenation event. Focusing on denitrification, 16S rRNA gene sequences does not provide information about which metabolic process evolved first: complete or partial denitrification. Loss of enzymes or nonacquisition of the complete pull enzymes are likely related to the adaptation to the environment and the electron acceptor availability. Intriguingly, NirK and Nor are the enzymes that have the higher co-occurrence, and they could be the key for denitrification evolution in haloarchaea. Therefore, their detoxification role should be addressed in future studies to establish their relationship with the pathway evolution.

## Figures and Tables

**Figure 1 microorganisms-09-01669-f001:**
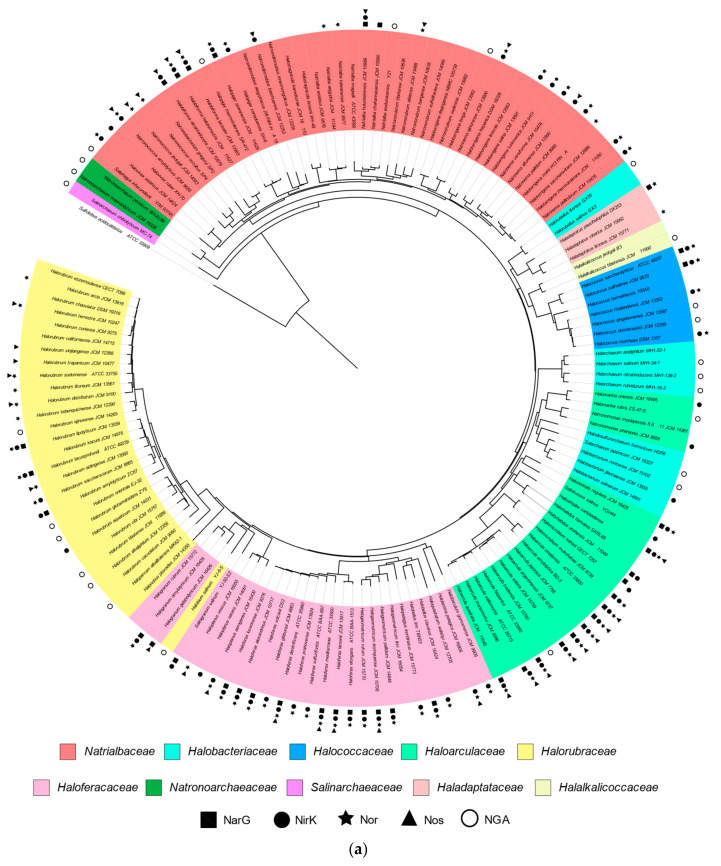
(**a**) Phylogenetic tree built with the 147 species of haloarchaea. Colour ranges indicate the family to which the microorganisms belong, and symbols specify which enzymes possess. NGA: no genome available. *Sulfolobus acidocaldarius* has been included as the root of the cladogram. (**b**) Distribution of denitrification enzymes among haloarchaea species indicating the number of species that present any of the four main enzymes of denitrification. ND: non-denitrifiers.

**Figure 2 microorganisms-09-01669-f002:**
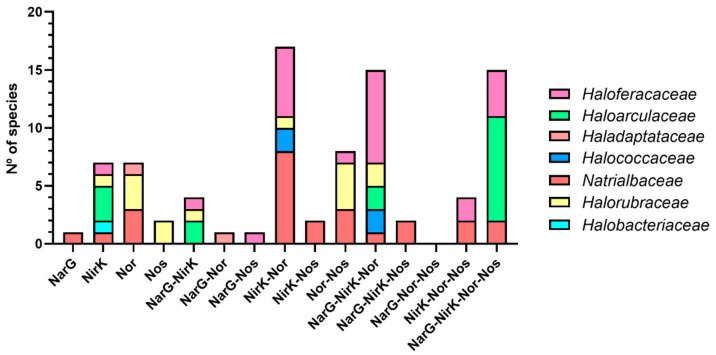
Co-occurrence of NarG, NirK, Nor and Nos in haloarchaea. Number of species in each family that present one, two, three or four genes encoded denitrification enzymes.

**Table 1 microorganisms-09-01669-t001:** Search equations employed in this work.

Gene or Protein	Search Equation
Respiratory Nitrate reductase (Nar)	((nitrate reductase) OR molybdopterin OR narg) AND *specie* [Organism]
Nitrite reductase (Nir)	((nitrite reductase) OR nirk OR nirs) AND *specie* [Organism]
Nitric oxide reductase (Nor)	((nitric oxide reductase) OR nor) AND *specie* [Organism]
Nitrous oxide reductase (Nos)	((nitrous oxide reductase) OR nos) AND *specie* [Organism]

**Table 2 microorganisms-09-01669-t002:** Classification of the species selected for this study by genera and family. In brackets the number of species that belong to that specific taxonomic level is displayed.

Families (10)		Genera (44)	
*Natrialbaceae* (44)	*Haloferacaceae* (27)	*Haladaptus* (3)	*Haloquadratum* (1)
*Halobacteriaceae* (11)	*Salinarchaceae* (1)	*Halalkalicoccus* (2)	*Halorhabdus* (2)
*Haloarculaceae* (21)	*Haladaptataceae* (3)	*Halarchaeum* (4)	*Halorientalis* (1)
*Halalkalicoccaceae* (2)	*Halorubraceae* (29)	*Haloarcula* (10)	*Halorubellus* (2)
*Natronoarchaceae* (2)	*Halococcaceae* (7)	*Halobacterium* (3)	*Halorubrum* (26)
		*Halobaculum* (1)	*Halosimplex* (1)
		*Halobellus* (2)	*Halostagnicola* (2)
		*Halobiforma* (3)	*Haloterrigena* (9)
		*Halobium* (1)	*Halovivax* (2)
		*Halococcus* (7)	*Natrialba* (7)
		*Halodesulfurarchaeum* (1)	*Natrinema* (5)
		*Haloferax* (10)	*Natronoarchaeum* (2)
		*Halogeometricum* (4)	*Natronobacterium* (1)
		*Halogranum* (3)	*Natronococcus* (3)
		*Haloamina* (1)	*Natronolimnobius* (3)
		*Halomarina* (2)	*Natronomonas* (2)
		*Halomicrobium* (2)	*Natronorubrum* (5)
		*Halonotius* (1)	*Salarchaeum* (1)
		*Haloparvum* (1)	*Salinarchaeum* (1)
		*Halopelagius* (1)	*Salinigranum* (1)
		*Halopiger* (3)	*Salinirussus* (1)
		*Haloplanus* (3)	*Saliphagus* (1)
		*Haladaptus* (3)	*Haloquadratum* (1)

**Table 3 microorganisms-09-01669-t003:** Detailed data by family. Percentages are exclusively calculated with species that present available genome in NCBI database.

Family	N° of Species	No Genome Available	Non-Denitrifiers (%)	Partial Denitrifiers (%)	Complete Denitrifiers (%)
*Natrialbaceae*	44	4	15 (37.5%)	23 (57.5%)	2 (5%)
*Halobacteriaceae*	11	6	4 (80%)	1 (20%)	-
*Halococcaceae*	7	2	1 (20%)	4 (80%)	-
*Haloarculaceae*	21	2	3 (15.8%)	7 (36.8%)	9 (47.4%)
*Halorubraceae*	29	7	8 (36.4%)	14 (63.6%)	-
*Haloferacaceae*	27	0	3 (11.1%)	20 (74.1%)	4 (14.8%)
*Natronoarchaceae*	2	2	-	-	-
*Salinarchaceae*	1	1	-	-	-
*Haladaptataceae*	3	0	1 (33.3%)	2 (66.7%)	-
*Halalkalicoccaceae*	2	1	1 (100%)	-	-
TOTAL	147	25	36 (29.5%)	71 (58.2%)	15 (12.3%)

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
