# Peer review of "Distribution of Denitrification among Haloarchaea: A Comprehensive Study"

_microorganisms, 2021, doi:10.3390/microorganisms9081669_

Round 1
Reviewer 1 Report
The authors present an analysis of archaeal genomes and identify the presence of genes involved in denitrification. The English is generally good although there are some minor errors. The methods and results are clear, but the paper is repetitive and relies on a limited amount of data. With some rigorous editing, this paper could support the conclusions drawn about the importance of denitrification in hypersaline systems (with the caveat that genomic data only indicates potential or capacity, not activity in situ).
Specific notes:
Table 2 - If genera are not going to be used in the analysis, why list them in the table? Also see Rinke et al, 2021, Nature Microbiology for an updated GTB.
Figure 1 - Natrialbaceae is in a different font. The white circles used for NGA are hard to see.
Figure 2 not necessary.
Table 3 - should be with figure 1.
Lines 238-274 - mostly repeating data in table 3.
Section 5.3 is speculation, and would need much more detailed genomic analysis to confirm statements about metabolic evolution. For example “high frequency of co-occurrence” (line 364) does not necessarily indicate that genes are linked.
There is also more repetition in lines 319-329. Figure 4 is unnecessary.
Author Response
The authors present an analysis of archaeal genomes and identify the presence of genes involved in denitrification. The English is generally good although there are some minor errors. The methods and results are clear, but the paper is repetitive and relies on a limited amount of data. With some rigorous editing, this paper could support the conclusions drawn about the importance of denitrification in hypersaline systems (with the caveat that genomic data only indicates potential or capacity, not activity in situ).
Thank you for your comments which contributes to the improvement of this MS. We have added a few lines (160-162) explaining that this genomic data only indicates potential capacity in Materials and Methods section. Moreover, some words were added to the text to clarify that denitrifiers in this study are “potential denitrifiers”, because with this study we cannot confirm if they can denitrify:
- Line 280: “potential”
- Line 317: “theoretical”
- Line 319: “potential”
- Line 320: “theoretically”
- Line 321: “potential”
- Line 322: “potential”
- Line 329: “potential”
- Line 438: “genes”
- Line 440: “potential”
In addition, some paragraphs have been rewritten to avoid reiterations
Specific notes:
Table 2 - If genera are not going to be used in the analysis, why list them in the table? Also see Rinke et al, 2021, Nature Microbiology for an updated GTB.
Thanks for your observation. Genera classification was added to the table for stating constancy that there were a low number of species for further genus analysis (First paragraph subsection 3.1). Reference 36 was updated to Rinke et al, 2021 (previously this reference was referred to the bacterial phylogenetic tree).
Figure 1 - Natrialbaceae is in a different font. The white circles used for NGA are hard to see.
Problems with font were fixed. White circles for NGA were modified and now their border is wider. Moreover, Figure 1 was provided separately to the editor at maximum quality to add it to the text and does not lose image quality.
Figure 2 not necessary.
Figure 2 displays part of the data that is presented in Figure 1. Thus, Figure 2 presents in a more visual way the number of species that presents denitrification enzymes, and we added this figure to make the text more easily readable and remark this information. To make Figure numeration more consistent Figure 2 is now Figure 1b and Figure 1 is now Figure 1a.
Table 3 - should be with figure 1.
Done.
Lines 238-274 - mostly repeating data in table 3.
Data from table 3 is discussed in these lines, linking denitrification with kind of environment. Text was modified to avoid the repetition of data that can be consulted in table 3.
Section 5.3 is speculation, and would need much more detailed genomic analysis to confirm statements about metabolic evolution. For example “high frequency of co-occurrence” (line 364) does not necessarily indicate that genes are linked.
Thanks for your comment. Lines 408-420 are speculative, and it is said in the conclusions that more research is needed to clarify the pathway evolution in haloarchaea (lines 439-440). About gene linkage it is important to said that these two genes not only have high co-occurrence, but also are close to each other in haloarchaeal genomes (see Ref. 25). This information was added to the text (line 398-399) and the word “seems to be” were also added before the word “linked” (line 398).
There is also more repetition in lines 319-329. Figure 4 is unnecessary.
We have rewritten this paragraph to avoid repetition.
Regarding Figure 4, we had represented the same information of Figure 3 in a more concise way to facilitate its interpretation, but as the information is the same, Figure 4 was removed.
Reviewer 2 Report
The study is focused on one of the most important biogeochemical cycles in nature, the N-cycle in haloarchaea inhabitating ecosystems with high salt concentration, high sun radiation and low availability of oxygen. Using phylogenetic methods, authors analysed comparative content of the N-cycle enzyme genes of the four main denitrifying enzyme complexes generally operating under microoxic/unaerobic conditions. The most valuable result of the study concerns of N2O- producing denitrifying enzyme which is a potent greenhouse gas and it contributes greatly to the depletion of the ozone layer. Authors showed that representatives of haloarchaeal families contain different sets of N-cycle enzymes. Since denitrification is not only the major source of biological N2O, but also the only known sink, this allows one to mitigate N2O emissions using haloarhaea and basing on knowledge of content and types of N-cycle enzymes inherent in haloarchaeal strains. The study is fulfilled with high professional level and can be interesting both to microbiologists and wide circles of scientists. Only few corrections should be made as listed in the attachment. The following are comments and suggested corrections to a few sentences with the following notations: [...] - for inclusion and ]...[ - for deletion:
Introduction:
43-53: This section is unclear and sometimes illogical, so it needs to be reformulated. I modified it and invite the authors to use it as a template.
Microbial communities play relevant roles in biogeochemical cycles in nature. One of them is the N-cycle, the conversion of nitrates into gaseous nitrogen-containing compounds and vice versa, which is of particular importance in connection with human anthropogenic activities. It is worth noting that anthropogenic activities such as agriculture contribute to the increase of nitrate, nitrite, and ammonium. This, in turn, leads to the fact that most of the nitrous oxide (N2O) is released by microorganisms after the transformation of the reactive nitrogen in fertilized agricultural zones [8–10]. N2O is a potent greenhouse gas and it contributes greatly to the depletion of the ozone layer [5–7]. In this regard, denitrification is noteworthy, because it is one of those metabolic pathways that is not only the major source of biological N2O, but also the only known sink [2], which allows one to mitigate N2O emissions. Therefore, microbes showing metabolic capabilities related to N-cycle are worth investigating.
Methods:
Is the following correct?
144: may ]are[ not [be] proper denitrifiers
Results:
204: of these sequences ]have no revealed different features[ [did not reveal any differences]
249: can be ]stablished[ [established]
257: many ]others[ [other] cycles
Author Response
Introduction:
43-53: This section is unclear and sometimes illogical, so it needs to be reformulated. I modified it and invite the authors to use it as a template.
Microbial communities play relevant roles in biogeochemical cycles in nature. One of them is the N-cycle, the conversion of nitrates into gaseous nitrogen-containing compounds and vice versa, which is of particular importance in connection with human anthropogenic activities. It is worth noting that anthropogenic activities such as agriculture contribute to the increase of nitrate, nitrite, and ammonium. This, in turn, leads to the fact that most of the nitrous oxide (N2O) is released by microorganisms after the transformation of the reactive nitrogen in fertilized agricultural zones [8–10]. N2O is a potent greenhouse gas and it contributes greatly to the depletion of the ozone layer [5–7]. In this regard, denitrification is noteworthy, because it is one of those metabolic pathways that is not only the major source of biological N2O, but also the only known sink [2], which allows one to mitigate N2O emissions. Therefore, microbes showing metabolic capabilities related to N-cycle are worth investigating.
Thank you for your kind correction. We have added the modified paragraph to the text and all the involved references were renumbered.
Methods:
Is the following correct?
144: may ]are[ not [be] proper denitrifiers
Modified as “may not be proper denitrifiers”
Results:
204: of these sequences ]have no revealed different features[ [did not reveal any differences]
249: can be ]stablished[ [established]
257: many ]others[ [other] cycles
The corrections have been made. Thank you very much.
Reviewer 3 Report
The submitted article by Miralles-Robledillo et al. contains a study of the denitrification activity of Halobateria. Although the idea of the analysis and the way it was carried out is interesting, several issues should be clarified.
1) It is not made clear on what basis organisms were selected for comparison. Were all Halobacteria from the database analyzed? If not then what was the criterion. A search by genes encoding nitroreductases is mentioned, but microorganisms lacking them are also mentioned.
2) Line 51-52 - This sentence is unclear. In what context do you use the word "sink"? As nitrification process?
3) I suggest that the manuscript should be classify as a review but not the common research article.
4) Line 163-164: You statet "The main objective of this research is not only to clarify the role of haloarchaea in denitrification, but also to deepen in their contribution to climate change due to the production of nitrogenous gases through denitrification." However, In my opinion the suggested aim of the study does not fix to the results. E.g. to evaluate the role of halobacteria in denitrification the conducted analysis of nitroreductase-coding genes existance is insufficient to make any such statements. Moreover, the conclusions about the impact on climate chage is deffinitely to far statement. There is a lack of other research and the lack of broader context of activity of other denitrifying and nitrifying microorganisms.
Author Response
1) It is not made clear on what basis organisms were selected for comparison. Were all Halobacteria from the database analyzed? If not then what was the criterion. A search by genes encoding nitroreductases is mentioned, but microorganisms lacking them are also mentioned.
RefSeq Database was consulted to recover all the available 16S rRNA gene sequences from haloarchaea (147 sequences/species). From these sequences, 122 species have an available genome (draft or complete) to seek for denitrification genes (this is mentioned at line 196-200).
Subsection 2.1 “Data sample and haloarchaeal phylogenetic tree constructing” was slightly modified to clarify how the 147 sequences were recovered to build the tree.
In this study, the comparisons have been made taking all species with available genome (122 genomes) into account. Then, the aim of the study was to explore distribution of denitrification among these species. The analysis reveals that not all the included microbes have nitroreductases. This is the reason explaining why some microbes are mentioned even if they lack those enzymes. This issue is mentioned in the text (for example see lines 318-328) when data (percentages) are given in terms of potential denitrifiers (only species carrying denitrification enzymes) or in terms of total available genomes (denitrifiers + non-denitrifiers).
2) Line 51-52 - This sentence is unclear. In what context do you use the word "sink"? As nitrification process?
The term “sink” is used to state that denitrification pathway is the only biological pathway in which the green-house gas N2O can be converted to N2. In other words, N2O can be “consumed” by microorganisms carrying nitrous oxide reductase, and these microorganisms are a sink of this gas. This term was used previously in literature with the same purpose (please see works from Prof. Lars Bakken, Asa Forstergard or Linda Bergaust among others about the use of the term “sink” in this context).
3) I suggest that the manuscript should be classify as a review but not the common research article.
Thank you for the suggestion. This manuscript is not evaluating previously published material, which is the nature of a proper Review. This is a new and original research based on bioinformatic analysis. There is a large genomic dataset retrieved, organized, and analyzed providing new results and insights about the importance of haloarchaeal denitrification. Due to the above mentioned, we firmly believe that this manuscript should be classified as a research article.
4) Line 163-164: You statet "The main objective of this research is not only to clarify the role of haloarchaea in denitrification, but also to deepen in their contribution to climate change due to the production of nitrogenous gases through denitrification." However, In my opinion the suggested aim of the study does not fix to the results. E.g. to evaluate the role of halobacteria in denitrification the conducted analysis of nitroreductase-coding genes existance is insufficient to make any such statements. Moreover, the conclusions about the impact on climate chage is deffinitely to far statement. There is a lack of other research and the lack of broader context of activity of other denitrifying and nitrifying microorganisms.
Thank you for your comment. Although in silico analysis, as the presented here, have some limitations, they constitute relevant studies to open and consolidate new research lines. Thus, these kinds of studies provide high valuable information being the basis for future in vivo/in vitro research. As the reviewer point out, the mere presence of genes in genomes may not be enough to strictly conclude that hypersaline environments could be sources of greenhouse gases such us NO or N2O. We agree that this fact must tested in the near future by biochemical/ecological studies. However, the high number of genomes analyzed, and the presence of the molecular machinery related to denitrification in many of them point to that way. Consequently, we state several times throughout the MS that the results obtained lead to new research for understanding the contribution of these ecosystems to climate change (examples: lines 339-343 and 443-446). Nevertheless, to be clearer about this “in vitro limitation” a sentence was added at lines 160-162 and some words were added to the text:
- Line 280: “potential”
- Line 317: “theoretical”
- Line 319: “potential”
- Line 320: “theoretically”
- Line 321: “potential”
- Line 322: “potential”
- Line 329: “potential”
- Line 438: “genes”
- Line 440: “potential”
Reviewer 4 Report
Review to the manuscript entitled “Distribution of denitrification among haloarchaea: A comprehensive study” by Jose María Miralles-Robledillo et al. submitted to Microorganisms (MDPI).
The manuscript focuses on denitrification in haloarchaea and phylogenetic approach was used for the study. Presence and abundance of different denitrification enzymes among Halobacteria were presented and thoroughly discussed. Therefore, the manuscript subject in accordance to the aims and scope of Microorganisms.
The in-depth study is of relevance to the scientific community working on archaea, subjects connected to N-cycle and on relevant enzymes. It could give relevant input to further ecological as well as molecular studies.
General remarks:
The manuscript is written in academic style and generally correct English language is used. Still, the wording can be improved in several sentences.
Some minor formatting details should be corrected.
Specific comments:
Lines 4, 5. Affiliations of all authors should be clarified.
Lines 27-29. 11 keywords are presented. According to the journal’s policy, up to 10 keywords should be given.
Lines 39-40. Gas vesicles are also widely known as gas vacuoles. The alternative name could be mentioned.
Line 46. Instead of “worthing” which is not in a correct form, another word should be used.
Lines 47-49. The sentence should be rewritten to enhance clarity.
Line 62. The meaning of “positive side” should be clarified.
Line 91. It is not clear what “in selection of” means in the sentence. Are authors referring to numerous or some selected bacterial species?
Line 109. Instead of “building”, the authors could use “constructing”.
Line 110, 115, 332, 408. The correct term for ribosomal rRNA sequences is “16S rRNA gene sequences”.
Line 119. The official nomenclature of prokaryotes should be used which provides the most accurate confirmed taxonomy. List of Prokaryotic names with Standing in Nomenclature can be found at the link: https://lpsn.dsmz.de/
Lines 139-146 and throughout the study. As explained in the section, in the study “denitrifier” is used if there is at least one gene encoding enzyme for denitrification. For clarity, these organisms should be named as microbes “with denitrification potential” as based only on the genomic data it is not possible to verify if the genes are expressed and the microbe can actually denitrify.
Lines 168-169. The date for the record should be mentioned as there are new genome sequences added to the databases frequently.
Figure 1. Due to the large dataset with high number of species mentioned, it is very difficult to read the text on the figure. For clarity and to ease the readers to look deeper into data presented on the figure the authors could provide magnified sections of the figure 1 (or different layout of the figure) in Supplementary Information.
Line 181. Instead of “codify”, “encode” should be used.
Lines 204-206. The sentence should be rewritten for clarity.
Lines 356-381. The authors could comment the similarities of haloarchaeal denitrification enzymes with that of halophilic bacteria, the potential source (or host) for horizontal gene transfer. It has to be noted that HGT is not very common process in archaea compared to bacteria.
Author Response
Lines 4, 5. Affiliations of all authors should be clarified.
Affiliations added to all authors.
Lines 27-29. 11 keywords are presented. According to the journal’s policy, up to 10 keywords should be given.
The keyword “phylogenetic distribution” was removed.
Lines 39-40. Gas vesicles are also widely known as gas vacuoles. The alternative name could be mentioned.
The term “gas vacuoles” was added (line 40)
Line 46. Instead of “worthing” which is not in a correct form, another word should be used.
Modified from “worthing to explore” to “worth to explore”.
Lines 47-49. The sentence should be rewritten to enhance clarity.
Sentence was modified.
Line 62. The meaning of “positive side” should be clarified.
The term “positive side” was clarified in the text.
Line 91. It is not clear what “in selection of” means in the sentence. Are authors referring to numerous or some selected bacterial species?
Corrected
Line 109. Instead of “building”, the authors could use “constructing”.
The word “building” was changed to “constructing”.
Line 110, 115, 332, 408. The correct term for ribosomal rRNA sequences is “16S rRNA gene sequences”.
The correct term was added.
Line 119. The official nomenclature of prokaryotes should be used which provides the most accurate confirmed taxonomy. List of Prokaryotic names with Standing in Nomenclature can be found at the link: https://lpsn.dsmz.de/
Thank you so much for you comment. We want to clarify that the used database in the study “Genome Taxonomy Database (GTDB) (https://gtdb.ecogenomic.org/)” integrate the data from LPSN for nomenclature and taxonomy.
Lines 139-146 and throughout the study. As explained in the section, in the study “denitrifier” is used if there is at least one gene encoding enzyme for denitrification. For clarity, these organisms should be named as microbes “with denitrification potential” as based only on the genomic data it is not possible to verify if the genes are expressed and the microbe can actually denitrify.
Thank you for the observation. We have added a few lines (160-162) explaining this limitation in Materials and Methods section. Moreover, some words were added to the text to clarify that denitrifiers in this study are “potential denitrifiers” because with this study we cannot confirm if they can denitrify:
- Line 280: “potential”
- Line 317: “theoretical”
- Line 319: “potential”
- Line 320: “theoretically”
- Line 321: “potential”
- Line 322: “potential”
- Line 329: “potential”
- Line 438: “genes”
- Line 440: “potential”
Lines 168-169. The date for the record should be mentioned as there are new genome sequences added to the databases frequently.
The date was added.
Figure 1. Due to the large dataset with high number of species mentioned, it is very difficult to read the text on the figure. For clarity and to ease the readers to look deeper into data presented on the figure the authors could provide magnified sections of the figure 1 (or different layout of the figure) in Supplementary Information.
I am afraid that images inserted directly to Microsoft Word are automatically compressed and this is noticeable with large pictures. To deal with this problem Figure 1 was provided separately to the editor at maximum quality to add it to the text and does not lose image quality.
Line 181. Instead of “codify”, “encode” should be used.
Modified.
Lines 204-206. The sentence should be rewritten for clarity.
Sentence was modified.
Lines 356-381. The authors could comment the similarities of haloarchaeal denitrification enzymes with that of halophilic bacteria, the potential source (or host) for horizontal gene transfer. It has to be noted that HGT is not very common process in archaea compared to bacteria.
Thank you for your comment. A brief explanation of the possible origin of these genes was added to the text (line 402-407).
Round 2
Reviewer 1 Report
N/A
Reviewer 3 Report
Thank you for the responses to my comments.